Contrasting molecular and morphological evidence for the identification of an anomalous Buteo: a cautionary tale for hybrid diagnosis

Clark William S. raptours@earthlink.net 1
Galen Spencer C. 2
Hull Joshua M. 3
Mayo Megan A. 3
Witt Christopher C. 4
1 Harlingen , TX , United States
2 Sackler Institute for Comparative Genomics, American Museum of Natural History , New York , NY , United States
3 Department of Animal Science, University of California , Davis , CA , United States
4 Museum of Southwestern Biology and Department of Biology, University of New Mexico , Albuquerque , NM , United States
Edwards Scott
Electronic publication date: 2017 Jan 10
Publication date: 2017
Volume: 5
Electronic Location ID: e2850
Received 2016 May 5; Accepted 2016 Dec 1
Copyright: ©2017 Clark et al.
Copyright year: 2017
Copyright holder: Clark et al.
License: This is an open access article distributed under the terms of the Creative Commons Attribution License, which permits unrestricted use, distribution, reproduction and adaptation in any medium and for any purpose provided that it is properly attributed. For attribution, the original author(s), title, publication source (PeerJ) and either DOI or URL of the article must be cited.
License URL: https://creativecommons.org/licenses/by/4.0/

Keywords: Unusual plumage, DNA, Phenology, Morphology, Hybrid, Specimen

Funding: NSF DEB-1146491 NSF GRF 1148897 Funding was provided in part by NSF DEB-1146491 and NSF GRF 1148897. The funders had no role in study design, data collection and analysis, decision to publish, or preparation of the manuscript.

==============================
An adult Buteo was found dead as a road-kill south of Sacramento, California, and was thought to represent the first state record of the eastern Red-shouldered Hawk (B. lineatus lineatus;). It is now a specimen in the Museum of Wildlife and Fisheries Biology (WFB 4816) at the University of California, Davis. We examined this specimen and found that many of its plumage characters differed from all other adult Red-shouldered Hawks examined, including nominate adults. Plumage markings and measurements were intermediate between Red-tailed Hawk (Buteo jamaicensis, ssp calurus) and Red-shouldered Hawk (ssp elegans), leading us to hypothesize that the bird was a hybrid. However, mtDNA sequences and nuDNA microsatellites proved definitively that the bird was a Red-shouldered Hawk, most likely of eastern origin. This case illustrates that apparent hybrids or apparent vagrants could be individuals with anomalous phenotypes caused by rare genetic variation or novel epigenetic effects.

Introduction

Hybridization often occurs between sympatric populations of closely related species, and it is a process that can have important evolutionary consequences (Seehausen, 2004; Rheindt & Edwards, 2011). Hybridization can accelerate or slow speciation (Abbott et al., 2013), can rescue inbred lineages (Frankham, 2015), and can lead to extinction (Allendorf et al., 2001). Often the youngest, or most recently radiated species experience hybridization (Mallet, 2005). Raptors of the genus Buteo are part of a rapid, recent radiation in which many species overlap in their distribution (Riesing et al., 2003). Several cases of hybridization among Buteo species have come to light relatively recently (e.g., Dudás, Tar & Tóth, 1999; Pfander & Schmigalew, 2001; Clark & Witt, 2006; Hull et al., 2007; Corso & Gildi, 1998; Gjershaug et al., 2010), often resulting in individuals with intermediate characteristics.

In this study, we examined an adult specimen that was similar in physical characteristics to adult Red-shouldered Hawks (B. lineatus), but with certain traits that were intermediate between Red-shouldered Hawk and Red-tailed Hawk (B. jamaicensis). This hybrid combination has not been previously noted in the peer-reviewed literature. This specimen provided an opportunity to use morphological and genetic data to more closely examine the possibility of hybridization between two species that are mobile and wide-ranging in North America. The specimen, found dead south of Sacramento, California, on 21 September 1996, had been described previously as a vagrant eastern Red-shouldered Hawk (B. l. lineatus), the first record for that taxon in California (Pyle et al., 2004). However, the possibility of hybrid status was not examined quantitatively, and no molecular data were used. We re-evaluated the findings of Pyle et al. (2004) for three reasons: (1) Many of the plumage markings and morphological measurements appeared to be different from Red-shouldered Hawk; (2) several recent records of hybrids between Buteo species suggest that hybridization is plausible; and (3) understanding circumstances where hybridization occurs can illuminate ongoing evolutionary processes within this genus.

Methods

We examined the plumage patterns and colors of 76 specimens of adult eastern and western Red-shouldered Hawks in nine museums (Appendix S1). We scored the following characters: malar and throat color, color and markings of the remiges and upper tail coverts, color of the lesser upper wing coverts, and the number of white tail bands. We compared these characters to those of the putative hybrid (WFB 4816). In an effort to find any eastern Red-shouldered Hawk individual that exhibited the plumage characters of WFB 4816, we examined more than 200 images of adult Red-shouldered Hawks in field guides and by using Google Image Search, and we examined photographs of 48 eastern adults captured for banding. After failing to find any Red-shouldered Hawks that resembled WFB 4816, we examined the 54 adult female specimens of eastern Red-shouldered Hawks in the US National Museum (USNM) that served as reference for Pyle et al. (2004).

We measured the wing chord, exposed culmen length, hallux chord, tarsus length, and standard tail length of adult specimens of three taxa: western Red-tailed Hawks (B. j. calurus) from California (n = 23), Red-shouldered Hawks from California (B. l. elegans; N = 33), and eastern North America (B. l. lineatus; N = 46), respectively. We centered and scaled the variables prior to conducting a principle components analysis (PCA) using the command prcomp in the stats package in R (http://www.r-project.org/). Individual measurements are shown in Appendix S1. After determining that the hallux chord measurement of WFB 4816 was larger than that of any Red-shouldered Hawk, we measured the halluces of additional specimens of adult females of B. l. lineatus from the Burke Museum (UWBM), Field Museum (FMNH), Museum of Vertebrate Zoology (MVZ), the Smithsonian Institution (USNM), and the Western Foundation of Vertebrate Zoology (WFVZ). This gave us measurements from a total of 98 adult female specimens of the nominate subspecies.

We extracted DNA from a toepad sample of WFB 4816 using a modified Qiagen DNeasy kit protocol by adding 30 ul 100 mg/ml DTT (dithiothreitol) to the proteinase K digestion and eluting the sample in 50 µl instead of the standard 200 µl. A negative control was included during the extraction process. We designed primers that amplify a 191 bp fragment of the mitochondrial gene ND2 (buteoND2F: CTGAACAAAATCCCCCACAC; buteoND2R: CGAGGCGGAGGTAGAAGAAT), and PCR amplified this fragment using 45 cycles of 94° for 30 s, 50° annealing for 30 s, and 72° extension for 1 min. We Sanger-sequenced this PCR product on an ABI 3130 machine.

Extracted DNA was amplified and genotyped for 18 microsatellite loci (A110, A302, A303, B111a, B220, B221, D107, D122, D123, D127, D207, D220, D223, D234, D235, D310, D327, D330) in six multiplex PCRs following the conditions described in Hull et al. (2007). PCR products were separated with a 3730 DNA Analyzer (Applied Biosystems Inc.), and then scored using STRAND version 2.3.89 (Toonen & Hughs, 2006). For comparison, we obtained genotypes from these loci for eastern and western Red-shouldered Hawks and Red-tailed Hawks from previous population genetic studies of these species (Hull et al., 2008a; Hull et al., 2008b).

To visualize the genetic relationships of the possible hybrid specimen with putative parental taxa, we performed a factorial correspondence analysis (FCA) (GENETIX 4.05.2; Belkhir et al., 2000). The FCA clustered genotypic groups based on microsatellite allele frequencies using the ordination of samples along varying factorial axes to visualize genetic similarity of the samples in two-dimensional space. In this analysis, a hybrid individual would be indicated by its intermediate location in the FCA between the genotypic groups for Red-shouldered and Red-tailed Hawks.

We also investigated the ancestry of the museum specimen through assignment tests using a Bayesian clustering algorithm (STRUCTURE version 2.1; Pritchard, Stephens & Donnelly, 2000). This analysis was first performed comparing the museum specimen with a combined eastern and western sample of Red-shouldered Hawks and a sample of Red-tailed Hawks. Subsequently, we performed a second analysis comparing the museum specimen to separate eastern Red-shouldered hawks and western Red-shouldered Hawks. No Red-tailed Hawks were included in the second test. For both analyses we used the population admixture model with a population prior corresponding to either species (test 1) or sampling location (test 2) and assumed that allele frequencies were correlated among populations. We ran the simulation with a 10,000 iteration burn-in and a run length of 100,000 iterations. We limited the analysis to K = 2 (where K is the number of populations) because of our prior knowledge of the species and geographic origin of the samples. We then generated Q plots to visualize proportion of ancestry for the museum specimen in relationship to possible populations of origin.

Results

The putative hybrid specimen (WFB 4816) differed in eight plumage characters from all adult Red-shouldered Hawks that we examined (130 specimens and >250 photographs for plumage and 79 specimens for measurements). Plumage differences are summarized in Table 1. Figure 1A shows that the putative hybrid has a dark throat, whereas typical adult Red-shouldered Hawks have pale throats with dark malar stripes. The putative hybrid also shows dark brown barring on the belly, a feature not exhibited by any adult Red-shouldered Hawk. Figure 1B shows that the putative hybrid has seven narrower white tail bands; in contrast, adult Red-shouldered Hawks usually have three or four wider bands (but a few had five bands). The tail bands and upper tail coverts on the putative hybrid show rufous tinges that are not seen on any adult Red-shouldered Hawk. Figure 1C shows the lack of white spotting on the uppersides of the remiges of the putative hybrid, whereas adult Red-shouldered Hawks always show numerous white spots. The lack of rufous on the lesser upper wing coverts of the putative hybrid is another characteristic that is inconsistent with Red-shouldered Hawk (shoulder; Fig. 1A). Many of the plumage differences listed above were also visible in Fig. 1 of Pyle et al. (2004).

Table 1 Comparison of plumage characters of the putative hybrid specimen with specimens of adult eastern and western Red-shouldered Hawks and western Red-tailed Hawks.

Character	WFB 4816	Adult East & West RS’s	Adult West RT’s	Figure no.	
Malar & throat	Uniformly dark	Darker malars; paler throat, often with dark streaks	Dark or dark with pale streaks	1	
Belly barring	Dark brown, with short dark streaks	Rufous	Dark brown with short dark streaks	1	
Uppersides of remiges	Gray barring, no white on primaries	Gray barring with white spots across primaries	Dark brown. No white on primaries	3	
Remige tips	Narrow gray tips	Wide white tips	Dark tips	3	
Lesser upper wing coverts	Dark brown with no rufous	Show rufous ‘shoulders’	Dark brown	3	
Upper tail coverts	Dark brown with rufous and white barring	Dark brown with white barring only	Rufous with dark brown barring	3	
No. of tail bands	6 (or 7)	4 or fewer (five had 5 bands)	No pale bands	2	
Tail band color	White with rufous tint	White only, no rufous	No pale bands	2	

Figure 1 Comparison of adult Eastern Red-shouldered Hawk (r) with WFB 4816 (l).

(A) Undersides. Note dark throat and dark belly markings of WFB 4816. (B) Upper tails. Note six tail bands of WFB 4816. (C) Upper wings. Note lack of white spotting on upper wings of WFB 4816.

Although the 54 adult female specimens of eastern Red-shouldered Hawks in the USNM collection had been consulted for the original identification of WFB 4816 as an eastern Red-shouldered Hawk, we found that none of these shared the combination of plumage traits exhibited WFB 4816. Seventeen of these USNM specimens showed five white tail bands, but none had seven as shown by WFB 4816. One had a mostly dark throat that was the same color as the malars, but it also had white streaks in the dark throat. Another specimen had gray markings on the upper wing coverts, but its remige tips and tail bands were white.

The largest hallux measurement on any adult female eastern Red-shouldered Hawk we measured was 23.9 mm (range 19.2–23.9 mm, x ¯=22.1, SD = 1.0, n = 98; Appendix S2); all were smaller than that of the putative hybrid specimen, 25.1 mm, which was more than two standard deviations greater than the mean of these measurements.

The PCA analysis of morphological measurements showed that eastern and western Red-shouldered Hawks each formed distinct clusters, but with some overlap. Red-tailed Hawks formed a separate, discrete cluster. The putative hybrid specimen fell outside of the ranges of measurement of any of the three taxa in our PCA (Fig. 2). Although it was closest to eastern Red-shouldered Hawks, it was also outside of the range of variation observed in other eastern Red-shouldered Hawks, and it was in between the centroids of California Red-shouldered Hawks and California Red-tailed Hawks on the PC1 axis, which explained 80.5% of the variance.

Figure 2 Plot of the first two principle components based on the PCA of five measurements from 103 adult eastern Red-shouldered Hawks (B. lineatus subspp.), California Red-shouldered Hawks (B. l. elegans), and western Red-tailed Hawks (B. j. calurus) representing both sexes showing that the putative hybrid is intermediate in its morphometric measurements between Red-shouldered Hawk and Red-tailed Hawk.

We successfully sequenced the complete 191 bp ND2 fragment of the specimen, WFB 4816 (GenBank accession number: KX154215) (Appendix S3). An NCBI BLAST search revealed that the sequence has 100% sequence identity to Red-shouldered Hawk (EU583276.1) and 99% similarity to two additional Red-shouldered Hawk sequences (GQ264875.1, GQ264874.1). All Red-tailed Hawk ND2 sequences on GenBank differ from WFB 4816 by at least 7 substitutions in the sequenced fragment (96% similarity).

Mitochondrial DNA (mtDNA) showed that the putative hybrid matches Red-shouldered Hawk, providing decisive evidence that its mother was a Red-shouldered Hawk. However, the mtDNA haplotype that we recovered was shared between eastern and western North American populations of Red-shouldered Hawk, so no geographic assignment could be made on the basis of mtDNA.

We recovered data from WFB 4816 for nine of the 18 microsatellite loci. Our failure to obtain data for the other nine loci was likely as a result of degraded DNA. The factorial correspondence analysis of the microsatellite data showed distinct clusters of individuals corresponding to Red-tailed Hawks and Red-shouldered Hawks in the comparison between these species (Fig. 3A), with WFB 4816 grouping within the Red-shouldered Hawk cluster. (Appendix S4 contains the raw microsat data). In the second factorial correspondence analysis, we found two clusters corresponding to eastern and western Red-shouldered Hawks with an area of overlap between the two (Fig. 3B). In this second analysis WFB 4816 grouped within the eastern Red-shouldered Hawk cluster.

Figure 3 Results of the factorial correspondence analysis (FCA) of microsatellite loci for WFB 4816 (yellow circles) compared to (A) eastern and western Red-shouldered Hawks, and (B) eastern and western Red-shouldered Hawks and western Red-tailed Hawks.

The Bayesian clustering analysis revealed a similar pattern to that seen in the factorial correspondence analysis. In the first analysis when K was set to 2 we found two distinct groups with very high proportions of ancestry corresponding to either Red-tailed Hawks or Red-shouldered Hawks (Fig. 4A). In this analysis, WFB 4816 showed ancestry solely from Red-shouldered Hawks. In the second analysis, the two a priori partitions (K = 2) corresponded to groupings of eastern and western Red-shouldered Hawks with low levels of admixture between the two groups (Fig. 4B). In this case, WFB 4816 showed a high proportion of ancestry derived from eastern Red-shouldered Hawks.

Figure 4 Results of Bayesian clustering analysis of microsatellite genotypes for (A) Red-tailed Hawks, eastern and western Red-shouldered Hawks, and WFB 4816 and (B) eastern and western Red-shouldered Hawks and WFB 4816.

(A) Compares the proportion of ancestry for each individual for K = 2 with one population composed of Red-shouldered Hawks (indicated in green) and on population of Red-tailed Hawks (indicated in red). (B) Compares the proportion of ancestry for each individual for K = 2 with one population composed of eastern Red-shouldered Hawks (indicated in green) and one population composed of western Red-shouldered Hawks (indicated in red); note that one of the Red-shouldered Hawks sampled in California appears to have mixed eastern and western ancestry. WFB 4816 is at the left of each figure and outlined in a dashed line. In (A) WFB 4816 shows a high proportion of Red-shouldered Hawk ancestry and in (B) WFB 4816 ii further indicated as having primarily eastern Red-shouldered Hawk ancestry.

Discussion

The microsatellite data for WFB 4816 provided a match with eastern group of Red-shouldered Hawks, corroborating the original conclusion of Pyle et al. (2004) that the specimen was a vagrant to California. Records of eastern Red-shouldered Hawks outside of their eastern range, though rare, have been recorded previously, including a juvenile specimen that was salvaged in Albuquerque, New Mexico, in 1991 (MSB:Bird:8174). The fact that nine microsatellite loci contained no evidence of mixed ancestry rules out the possibility that the specimen represents an hybrid between eastern Red-shouldered Hawk and Red-tailed Hawk (F1) because such a hybrid would have nine microsatellite alleles of Red-tailed Hawk origin. A first generation backcross (BC1) with eastern Red-shouldered Hawk would be expected to retain four-and-a-half Red-tailed Hawk alleles, on average, so would most likely have been detectable by our data. At the second backcross generation (BC2), an average of two-and-a-quarter Red-tailed Hawk alleles would be expected in our data, but the stochastic nature of Mendelian segregation makes it reasonably probable that our microsatellite data would fail to detect Red-tailed Hawk ancestry, even while one-eighth of the genome would be of Red-tailed Hawk origin. At subsequent backcross generations with eastern Red-shouldered Hawk, the genetic contribution of Red-tailed Hawk would certainly be too dilute to detect with a data set of this size (Lavretsky et al., 2016).

In contrast to our microsatellite data, the results of our analyses of phenotype were inconsistent with the conclusion of Pyle et al. (2004) that the specimen represented a eastern Red-shouldered Hawk. The morphometric measurements and plumage traits of WFB 4816 were atypical for Red-shouldered Hawk and appeared to be at least somewhat intermediate between Red-tailed Hawk and Red-shouldered Hawk, suggesting that the specimen represented a hybrid or a subsequent backcross with Red-shouldered Hawk. The first principle component of the morphological measurements (PC1), which provides an index of overall size (Wright, Steadman & Witt, 2016), showed that this specimen was larger than any of the 46 eastern Red-shouldered Hawk specimens for which we measured all five traits. Furthermore, its combination of plumage markings did not occur in any specimen or photograph that we examined and resembled an amalgamation of characteristics from both species.

How can we reconcile the apparent conflict between our genetic and phenotypic results? Hybrid individuals (F1) are typically intermediate in size and plumage between the parental species (Rohwer, 1994), but our microsatellite results preclude the possibility that WFB 4816 represents an F1 and they cast doubt on the possibility of a BC1 backcross with eastern Red-shouldered Hawk. However, our microsatellite results clearly cannot rule out a backcross at generation BC2 or higher. Thus, we cannot rule out the possibility that a Red-tailed Hawk contribution of one-eighth or less of the genome may have caused the unusual phenotype of WFB 4816, after two or more generations of backcrossing with eastern Red-shouldered Hawks. It is important to note that at each backcross generation, offspring are expected to increasingly resemble the backcrossed parental species due to exponential dilution of hybrid-origin alleles that underlie the phenotype. Consistent with the expectation under hybridization and subsequent backcrossing, the phenotype of WFB 4816 appears to be more similar overall to Red-shouldered Hawk than to Red-tailed Hawk (Figs. 1 and 2 and Table 1). Thus, hybridization followed by two generations of backcrossing is a viable explanation for WFB 4816, but additional genetic data would be required to test this hypothesis.

Occasional hybridization may result in reciprocal exchange of alleles between species if resultant offspring can successfully backcross, and particularly if an allele confers a selective advantage to the recipient species (Rheindt & Edwards, 2011). The plausibility of Red-tailed Hawk origin for the atypical characteristics of WFB 4816, whether by adaptive introgression or hybridization followed by two or more generations of backcrossing, is supported by a growing set of documented examples of hybridization involving species in the genus Buteo. A putative juvenile hybrid specimen in the collection of the Cleveland Museum of Natural History (catalog no. 69422) was banded in the nest near Cleveland, Ohio, and later found dead in the nearby area. Mother was a Red-tailed Hawk, and father was a Red-shouldered Hawk. Measurements and plumage traits are inconsistent with juveniles of either species. Photographs of an adult hybrid between Red-tailed Hawks and Red-shouldered Hawk from Connecticut have been posted online (Connecticut Audubon Society, 2011). Additional Buteo hybrids include those reported by Clark & Witt (2006), a hybrid specimen of Rough-legged (B. lagopus) x Swainson’s (B. swainsoni) Hawks, and Hull et al. (2007) on several cases of hybrids between Red-tailed and Swainson’s Hawks. Both of these papers also used DNA, plumages, and morphometrics to determine hybridization. We have examined unpublished photographs of two adults that show characters of two species, one of Red-tailed and Ferruginous (B. regalis) Hawks and the other of Red-tailed and Rough-legged Hawks. Four further hybridizations between two species of Buteo have been reported from Eurasia—Rough-legged x Common (B. buteo) Buzzards (Gjershaug et al., 2010), Long-legged x Common Buzzards (Dudás, Tar & Tóth, 1999; Corso, 2007), and Long-legged x Upland (B. hemilasius) Buzzards (Pfander & Schmigalew, 2001). Clark, Reid & BK (2005) reported on four instances of hybrids, three between species in Buteo, but also an intergeneric case of Swainson’s Hawk male and Red-backed Buzzard (Geranoaetus polyosoma) in Colorado (Allen, 1988). The most phylogenetically distant hybrid pairing involving a Buteo is that reported by Corso & Gildi (1998) from Italy of a Common Buzzard mated with a Black Kite (Milvus migrans) that produced two fledglings.

There are at least two alternative explanations for the anomalous phenotype of WFB 4816 that would not involve a genetic contribution from Red-tailed Hawk. First, the bird may have been subject to epigenetic, developmental abnormalities such as those that can be caused by environmental factors (Wong, Gottesman & Petronis, 2005). Second, an allele of large effect (Thompson & Jiggins, 2014) may have arisen via mutation or may be segregating in the population at such low frequency that it was previously undetected.

On the basis of our results, we conclude that WFB 4816 is an eastern Red-shouldered Hawk that is both a vagrant to California and an individual with an anomalous phenotype of uncertain cause. The confluence of these two unlikely events suggests that they are linked, and it is plausible that a rare allele of large effect, hybridization with Red-tailed Hawk two or more generations earlier, or an epigenetic developmental abnormality could have jointly affected the phenotype and the direction of migratory orientation. The compass direction of migratory orientation is generally subject to genetic control (Helbig, 1996).

Hybrid diagnoses that fail to obtain definitive genetic evidence of hybridization should consider at least three alternative explanations for phenotypic character states that are apparently intermediate between two parental species: (1) rare alleles of large effect that originated via mutation; (2) alleles of hybrid origin that persisted or introgressed after backcrossing, or (3) epigenetic developmental abnormalities. As whole-genome sequencing from museum specimens becomes increasing feasible (McCormack, Tsai & Faircloth, 2015), we will be better able to empirically distinguish among these potential causes of unusual and anomalous phenotypes.

Supplemental Information

Appendix S1 Morphometric data used in the PCA

Click here for additional data file.

Appendix S2 Measurements of hallux chord from 98 adult female B. l. lineatus specimens

Click here for additional data file.

Appendix S3 ND2 sequence and GenBank accession number obtained from voucher specimen WFB 4816

Click here for additional data file.

Appendix S4 Raw microsat data

Click here for additional data file.

We would like to thank the American Museum of Natural History, the Museum of Southwestern Biology of the University of New Mexico, the US National Museum, the Western Foundation of Vertebrate Zoology, and the Museum of Wildlife and Fish Biology of the University of California-Davis for making their collections open to our study.

Additional Information and Declarations

Competing Interests

Author Contributions

DNA Deposition

Data Availability

The authors declare there are no competing interests.

William S. Clark, Spencer C. Galen, Joshua M. Hull, Megan A. Mayo and Christopher C. Witt conceived and designed the experiments, performed the experiments, analyzed the data, contributed reagents/materials/analysis tools, wrote the paper, prepared figures and/or tables, reviewed drafts of the paper.

The following information was supplied regarding the deposition of DNA sequences:

GenBank accession number: KX154215.

The following information was supplied regarding data availability:

The raw data has been supplied as a Supplemental Dataset.

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
