# Peer review of "Contrasting molecular and morphological evidence for the identification of an anomalous Buteo: a cautionary tale for hybrid diagnosis"

_PeerJ, doi:10.7717/peerj.2850_

## Round 0.1 · original submission · Major Revisions

· Academic Editor

Major Revisions

I have received two thorough reviews of your paper and both reviewers have substantive comments to improve the manuscript. Reviewer 1 suggested caution in ruling out an advanced backcross bird, whereas reviewer 2 was concerned with the overall flow of logic, and the perhaps unjustified urge to call this bird a hybrid. I realize that these two assessments are somewhat contradictory but, as reviewer 1 notes, who knows what light advanced sequencing techniques would shed on this issue. Or perhaps you feel that your genetic analysis already rules out some of the possibilities raised by the reviewers. In general, both reviewers are very clear about the need to temper several statements and to clarify how this paper is an advance over Pyle (2004).

Reviewer 1 ·

Basic reporting

All comments are included in General Comments for the Author.

Experimental design

All comments are included in General Comments for the Author.

Validity of the findings

All comments are included in General Comments for the Author.

Additional comments

PeerJ #10497 on an anomalous Buteo.

The authors address a specimen from California that was thought to be the first state record of the eastern Red-shouldered Hawk, Buteo l. lineatus. Plumage and measurements suggested a possible hybrid between Red-tailed and Red-shouldered hawks. Genetic evidence, however, showed that it was just a phenotypically atypical Red-shouldered Hawk.

The only problem I have with the manuscript is that you do not have the power to rule out hybridization as a cause for the bird’s odd phenotype. Developmental anomalies in birds are not uncommon, but not in suites of characters as you describe and illustrate here. So in the paragraph beginning on line 167 I’d be a lot more cautious. Lavretsky et al. 2016 (Molecular Ecology 25:661-674) recently modeled how quickly genomic evidence of hybridization is lost in backcrossing. With so few loci, such a loss in your dataset would likely be faster, perhaps F2-3 (you could model it if you wanted using your data). You may also want to look at some of Graves’ detailed analyses of hybridization. Two in particular seem relevant: Proceedings of the Biological Society of Washington 1990 (103:6-25) and 1996 (109:373-390). Although they lack a genetic component, there is good discussion of hybrid phenotypic characteristics that is too often overlooked. I think your bird is probably a hybrid backcrossed several generations with Red-shouldered. Come back to it again in a few years when genomes are bit more cost effective and I’ll bet there will be some Red-tailed alleles in there.

Minor issues:

Line 48: Please give the date and locality of the specimen.

Line 65: Can you be clearer on the procedure of “We centered and scaled the variables...”? Log transformation seems like a more common procedure.

Lines 80-82: Provide sample sizes in parentheses after each named taxon. I am concerned that it looks like microsatellite genotypes were taken from the literature and that the taxa were not compared in direct, simultaneous runs on the same instrument using the same protocols to ensure accurate genotyping.

Lines 94-95: Unclear here – a) are these the same sample sizes from 80-82; and b) usually STRUCTURE is run without assigning individuals to groups, and that is how this analysis should be run first – it would give the reader confidence that the molecular dataset is sufficiently strong to distinguish the possible parental taxa. If this was done already by the Hull 2008a & b papers, give a summary of that so we are clear that these data provide definitive assignments for the parental taxa.

Line 101: You ran two analyses. Both at K = 2?

Lines 196-197: No – the situation is much more complex. Novel genes can introgress if the fitness of the hybrid/backcrossed individual is equal to or greater than the parental form (not exclusively if a selective advantage is present, as stated). And neutral changes can percolate quite effectively, and these represent evolutionary changes as well.

Lines 176-187: For all these pairs, Hawks and Buzzards should be lower-cased hawks and buzzards.

Line 297: I don’t see hallux in Table 1.

·

Basic reporting

A hybrid from a unique pairing in central California of a female Common Black-Hawk and a male Red-shouldered Hawk has been documented and the paper is currently under review for “Western Birds” but will soon be “in press” and could be cited by the authors. I can supply the manuscript to the authors if requested.

The introduction provides background to support the well-defined hypothesis. The authors seem to overstate the importance of their investigation with the statement that “hybridization is regularly occurring in the genus”. I am not sure what “regularly occurring” means, as hybridization in Buteo seems to be a rare occurrence based upon very few documented cases. With the thousands of birders and raptor biologists scrutinizing the identification of countless Buteos in North America, I’d suspect that many more cases would be documented if hybrids were “regularly occurring” enough to “illuminate the ongoing evolutionary processes within this genus”.

“107 The putative hybrid specimen (WFB 4816) differed in plumage from all adult Red-shouldered
108 Hawks, and, specifically, from adult eastern ones.” This sweeping statement should be revised to state that it differed from all specimens analyzed for this manuscript. They should also temper their statements to state that it differed from typical individuals of the species—not all individuals (need to keep open the possibility of oddly plumaged birds in the population).

Experimental design

No comments.

Validity of the findings

The instigating premise of the paper is a refutation of Pyle et al.’s (2004) conclusion that the specimen was of a Buteo linneatus linneatus. The authors of the current paper dismiss this conclusion without critically evaluating that initial conclusion and its supporting claims. They leap into their hybrid hypothesis yet circle back to accept the original conclusion based upon genetic results yet do not offer much explanation for how they were misled into believing this was a hybrid nor do they adequately rectify the disconnect between Pyle et al.’s (2004) conclusion and their initial rejection of that conclusion. After all, Pyle et al. (2004) stated that the plumage characters were diagnostic of B. l. linneatus. The fact that Pyle et al. (2004) analyzed 183 specimens vs. 76 in this manuscript is not mentioned. Perhaps some specimens evaluated by Pyle et al. (2004) fit the plumage of this featured specimen? Some readers may wonder if some of the “anomalous” plumage characters could be explained by the possibility of this specimen being a second-year female. The authors should at least address and discount, if necessary, this possibility. Also the authors state that this specimen is too large for a B. l. linneatus, yet all measurements, except for the hallux, fit well within the range of females in the specimens that they examined and conform to Pyle et al.’s (2004) results as well, so it seems a bit disingenuous to state that the specimen’s measurements “comprise compelling evidence” that it was a hybrid.

---

## Round 0.2 · accepted · Accept

· Academic Editor

Accept

Thanks for addressing all the reviewers' comments.